Methods

# Automated staging of zebrafish embryos with deep learning

Rebecca A Jones[1,2] , Matthew J Renshaw[3], David J Barry[3]

**The zebrafish (*Danio rerio*) is an important biomedical model organism used in many disciplines. The phenomenon of developmental delay in zebrafish embryos has been widely reported as part of a mutant or treatment-induced phenotype. However, the detection and quantification of these delays is often achieved through manual observation, which is both time-consuming and subjective. We present KimmelNet, a deep learning model trained to predict embryo age (hours post fertilisation) from 2D brightfield images. KimmelNet's predictions agree closely with established staging methods and can detect developmental delays between populations with high confidence using as few as 100 images. Moreover, KimmelNet generalises to previously unseen data, with transfer learning enhancing its performance. With the ability to analyse tens of thousands of standard brightfield microscopy images on a timescale of minutes, we envisage that KimmelNet will be a valuable resource for the developmental biology community. Furthermore, the approach we have used could easily be adapted to generate models for other organisms.**

## Introduction

The zebrafish (*Danio rerio*) is a commonly-used model organism in the field of developmental biology, with their transparent embryos making them particularly conducive to microscopy observation. Many developmental studies require accurate staging of zebrafish embryos and larvae. Although the timing of fertilisation can be estimated to within ~30 min, the hours post fertilisation (hpf) at the standard temperature of 28.5°C provides only an approximation of the actual developmental stage because other factors, like population density and water quality, can affect maturation (Singleman & Holtzman, 2014). Even controlling for such factors, embryos within a clutch may develop at different rates (Parichy et al, 2009). Researchers therefore use both hpf and staging guides that are based on morphological criteria to stage individual embryos. These morphological features include the number of somites and the appearance of landmark structures such as the embryonic shield, tail bud, and eye primordium (Kimmel et al, 1995; Westerfield, 1995).

Staging of embryos is of particular importance because many studies report "developmental delay" as part of a genetic or drug-induced phenotype (Giraldez et al, 2005; Akthar et al, 2018; Byrnes et al, 2018; Elabd et al, 2019; Farooq et al, 2019; Jia et al, 2020). Zebrafish have also emerged as important models in which to study the effects of environmental toxins, with many of these treatments also resulting in developmental delay (Mesquita et al, 2017; Aksakal & Sisman, 2020; Li et al, 2020). Such delays are difficult to quantify without manually staging large numbers of embryos, which is subjective and time-consuming (Jeanray et al, 2015; Teixido et al, 2019). A more standardized and automated approach to assessing developmental delay is therefore desirable, in order to avoid the substantial time cost and subjectivity associated with manual staging.

The use of image analysis has become increasingly popular in the life sciences in recent years, facilitating the automated quantification of microscopy images in an unbiased fashion (Meijering et al, 2016). But, designing an image analysis algorithm to specifically detect the wide range of morphological features on which staging guides depend would be a challenging endeavour. However, the staging of embryos based on microscopy images could also be considered an image classification task, something to which machine learning is well-suited. Machine learning approaches, where a computer program uses algorithms and statistical models to continuously learn and improve pattern prediction, are already used widely in biological studies (Greener et al, 2022).

Machine learning has found some use in the analysis of zebrafish embryos, but mainly for phenotype classification, rather than assessing developmental delay. For example, using a support vector machine, Liu et al (2012) achieved a classification accuracy of ~97% in differentiating between hatched, unhatched, and dead zebrafish embryos. Jeanray et al (2015) used a supervised machine learning approach to place brightfield images of 3-d-old zebrafish larvae in 1 of 11 phenotypic classes. But their method involved significant image pre-processing and their classifier failed to accurately classify certain phenotypes.

[1]Department of Molecular Biology, Princeton University, Princeton, NJ, USA  [2]Developmental Biology Laboratory, The Francis Crick Institute, London, UK  [3]Crick Advanced Light Microscopy (CALM), The Francis Crick Institute, London, UK

Correspondence: david.barry@crick.ac.uk

More recently, convolutional neural networks (CNNs) have grown in popularity in life science research in general, and they are particularly well-suited to the analysis of image data (Greener et al, 2022). There are a number of examples in the literature of CNNs being used for zebrafish embryo phenotype classification. Ishaq et al (2017) used a CNN-based approach to predict the probability of deformation in multi-fish microwell plates. Although few images were needed for training (<100) and an F score of 0.915 was achieved, their network could only differentiate between two different classes. Tyagi et al (2018) tested multiple CNN architectures to classify zebrafish embryo phenotypes but achieved a relatively low accuracy (83.55%) when attempting to classify a large number of phenotypes (11). Using the image data from Jeanray et al (2015), Shang et al (2020) achieved a higher accuracy using a two-tier CNN-based approach, but again, the number of images used for training (529) and testing (341) across 10 classes was limited.

In terms of developmental staging, Pond et al (2021) developed a CNN to stage zebrafish tail buds at four discrete developmental stages just 1.5 h apart. Although they trained their network with a small number of images (<100) and achieved 100% test accuracy in some cases, the number of test images per class was in several cases as low as 2.

We recently published a machine learning-based method for the staging of zebrafish embryos (Jones et al, 2022). Although this method successfully discriminated between different populations of zebrafish developing at different rates, the developmental profiles it produced were distinctly non-linear. This was largely because of the fact that the underlying classifier was trained to recognise two distinct classes corresponding to two different stages of development (embryos at 4.5 and 17.5 hpf). As such, it performed poorly at accurately staging embryos in between those specific time points, producing a developmental profile with a distinctive "S" shape.

Here, we present a more robust classifier, based on a CNN, that provides more accurate predictions of developmental stage across a much wider range of timepoints (4.5 to 51 hpf). Named KimmelNet and using simple brightfield microscopy images as training data (with no manual annotation required), the developmental profiles produced by our CNN closely agree with those predicted by established approaches (Kimmel et al, 1995). We demonstrate that KimmelNet accurately quantifies developmental delay between two populations of zebrafish embryos. We also demonstrate that KimmelNet can successfully distinguish between populations in previously "unseen" data (that is, completely distinct from the training data) and this performance can be improved significantly with modest levels of transfer learning. We envisage that our classifier will become a valuable resource for the zebrafish community.

## Results and Discussion

### Network architecture

The architecture of KimmelNet is based on that described by Ishaq et al (2017), which is in turn a simplified version of AlexNet (Krizhevsky et al, 2012). To reduce the number of trainable parameters (and change the nature of the network from classification to regression), we simplified the architecture further still (Fig 1). The architecture consists of repeating units of three layers, consisting of 3 × 3 2D convolution, a rectified linear unit and 2 × 2 max pooling. The output consists of a single fully connected layer, a dropout layer, and a final fully connected (dense) layer consisting of a single output neuron. The total number of repeating convolution units, the number of convolutional filters in each and the dropout rate in the second-to-last layer were all determined using hyperparameter tuning, implemented using KerasTuner (Chollet, 2015).

### Training KimmelNet

KimmelNet was trained on 23,370 individual images of zebrafish embryos grown at 28.5°C, ranging in age from 4.5 to 51.75 hpf (Fig 2A and Table 1). We trained KimmelNet for 500 epochs as, for a greater number of iterations, the validation loss began to plateau (whereas the training loss continued to decrease slightly), indicative of over-fitting. After 500 epochs, training loss was ~8.5 (Fig 2B). Training took ~2.5 h on the high performance computing (HPC) cluster at the Francis Crick Institute (see the Materials and Methods section for hardware specifications).

### KimmelNet detects developmental delay with high confidence

We evaluated KimmelNet on two test datasets, one of zebrafish embryos grown at 28.5°C and one at 25.0°C (datasets "A" and "B" in Table 1). Testing KimmelNet on dataset A took ~160 s on the Crick's HPC hardware (see the Materials and Methods section), equating to just under 7 milliseconds per image.

To evaluate the accuracy of predicted hpf values, we fit an equation of the form $y = mx$ to predictions produced by KimmelNet. We used an equation of this form in order to be consistent with that used by Kimmel et al (1995):

$$H_T = \frac{h}{0.055T - c} \tag{1}$$

where $H_T$ corresponds to hours of development at temperature $T$ (°C), $h$ denotes the number of hours required to reach the equivalent developmental stage at 28.5°C, and $c$ is a constant equal to 0.57°C.

Fitting an equation of the form $y = mx$ to dataset A predictions resulted in a line with a slope equal to 0.974 ($R^2 = 0.951$) – a slope of 1.0 would indicate a perfect 1-to-1 correspondence between true hpf labels and predicted values (Fig 3A). Saliency maps to illustrate pixel attribution, generated using an implementation of DeconvNet (Zeiler & Fergus, 2013 Preprint), indicated that network predictions were driven by morphological features of the embryos, rather than any irrelevant features of the image background (Fig 4).

For dataset B (Table 1), Equation (1) predicts that embryos will develop at ~80.5% of the rate of those grown at 28.5°C, equivalent to a straight line with slope 0.805. A straight line fit to the predictions produced by KimmelNet reveal a lower rate of development (slope = 0.725, $R^2 = 0.949$; Fig 3A). However, the straight line

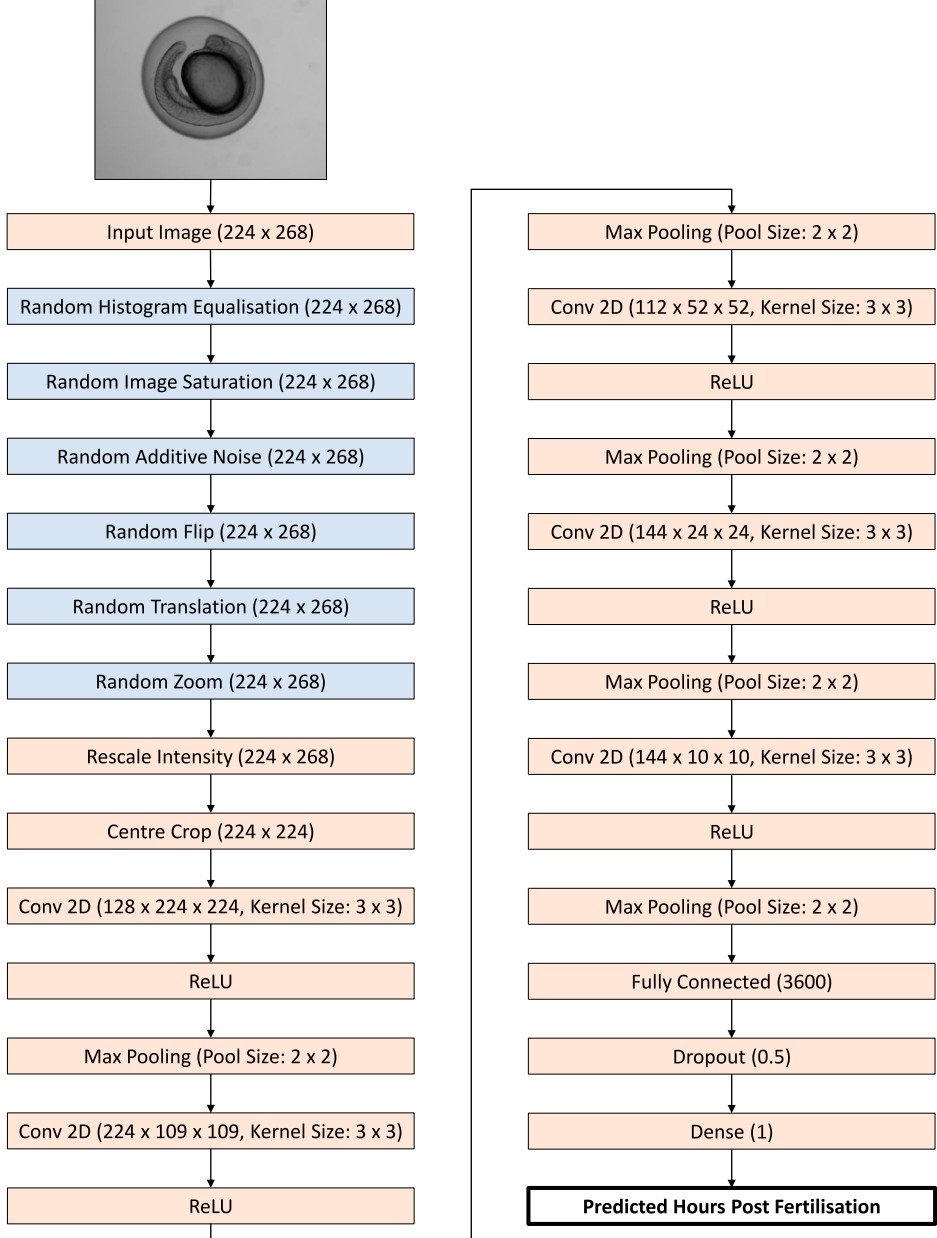

**Figure 1. Illustration of the KimmelNet architecture.**
Augmentation layers active for training only are shown in blue.

corresponding to Equation (1) falls within the range of predictions produced by KimmelNet (Fig 3A).

To determine how KimmelNet would perform on smaller test datasets, we took random samples of the data points shown in Fig 3A and fit lines to these points. Performing this sampling 10,000 times for either 100 or 200 random data points produced the range of line fits shown in Fig 3B. From ~2 hpf, there is no overlap between the confidence intervals for the two test datasets. This demonstrates that KimmelNet can reliably distinguish between two populations of zebrafish embryos developing at different rates given just 100 images of each.

Plotting the residuals to the straight line fits showed that the mean prediction errors for both test datasets were distributed approximately uniformly about zero (Fig 3C and D). For dataset A, the mean prediction error was 0.364 ± 0.019 h (mean ± s.e.m.) and the SD was 2.824 h. For dataset B, the mean prediction error was 0.302 ± 0.017 h and the SD was 2.136 h.

It is possible that the use of a more sophisticated model, or modification of the approach to model training, could result in a narrower distribution of prediction errors. However, it is also possible that these errors result from the approach used for annotating the training data. Each image in the training dataset is annotated with its associated hpf. However, it is known that

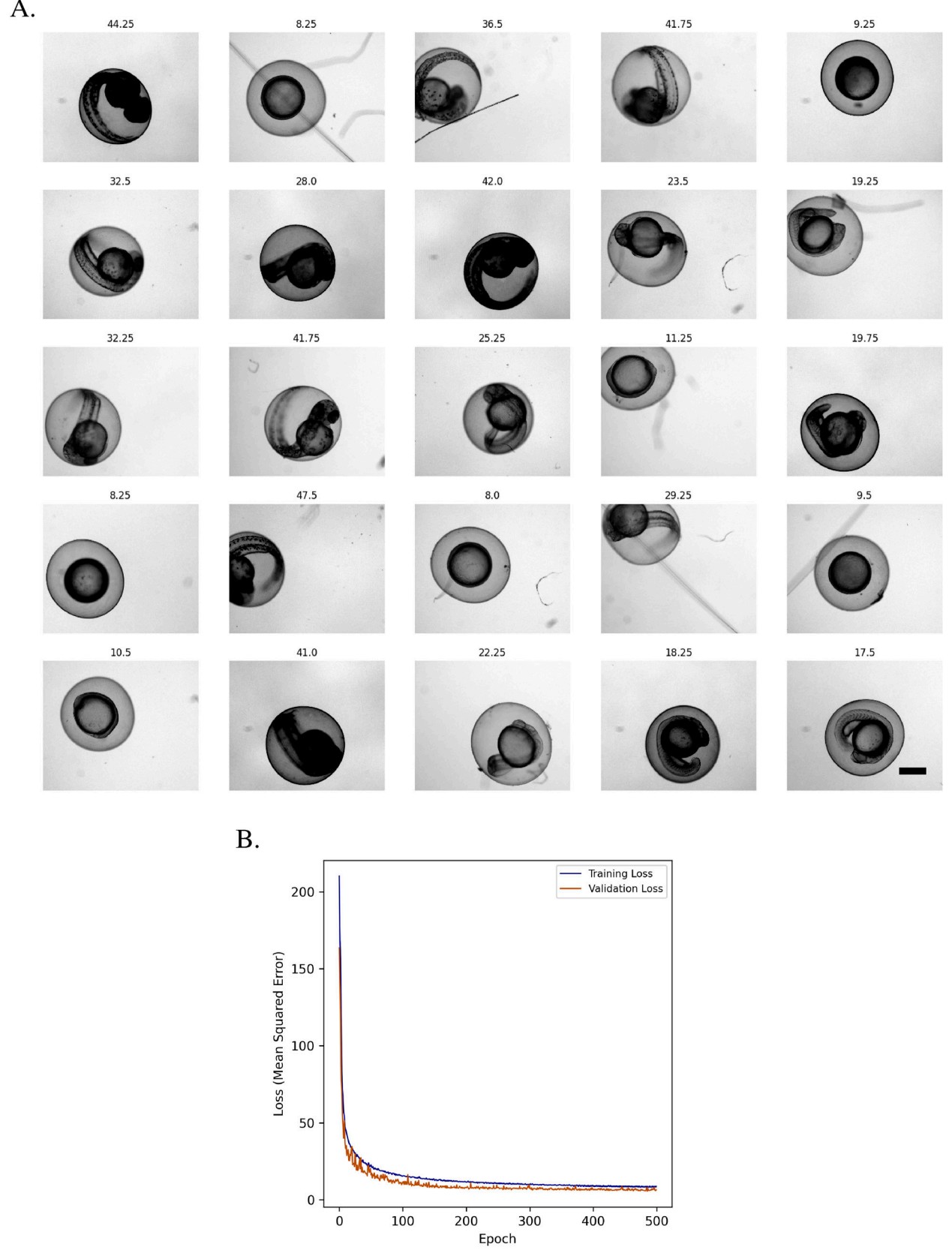

**Figure 2. KimmelNet converged to a training loss of ~8.5 after 500 epochs.**
**(A)** Example images of zebrafish embryos at various indicated stages of development—labels indicate hours post-fertilisation. Images were drawn from randomly selected wells in two different multi-well plates (datasets "A" and "B" in Table 1). Scale bar is 500 μm and all images are scaled equally. **(B)** Plot of training and validation loss during training of KimmelNet with ~23,000 images similar to those shown in "A."

**Table 1. Overview of datasets used for training and testing.**

| Dataset | BioImage archive accession no. | Imaging location | Incubation temp. (°C) | Number of plates | Number of images | % images used for training | % images used for testing | Prediction errors (mean ± sd) | Slope of line fit to predictions | R² of line fit to predictions |
|---|---|---|---|---|---|---|---|---|---|---|
| A | S-BIAD531 | Crick | 28.5 | 3 | 50,730 | 46 | 54 | 0.36 ± 2.82 | 0.974 | 0.951 |
| B | S-BIAD531 | Crick | 25 | 1 | 18,050 | 0 | 100 | 0.30 ± 2.14 | 0.725 | 0.949 |
| C | S-BIAD840 | Princeton | 28.5 | 1 | 16,512 | 0 | 100 | 1.30 ± 5.52 | 0.923 | 0.644 |
| D | S-BIAD840 | Princeton | 25 | 1 | 16,896 | 0 | 100 | 1.18 ± 5.82 | 0.773 | 0.533 |

Datasets A and B are the same as those used in a previously published study (Jones et al, 2022). All data are available to download from the BioImage Archive (Hartley et al, 2022).

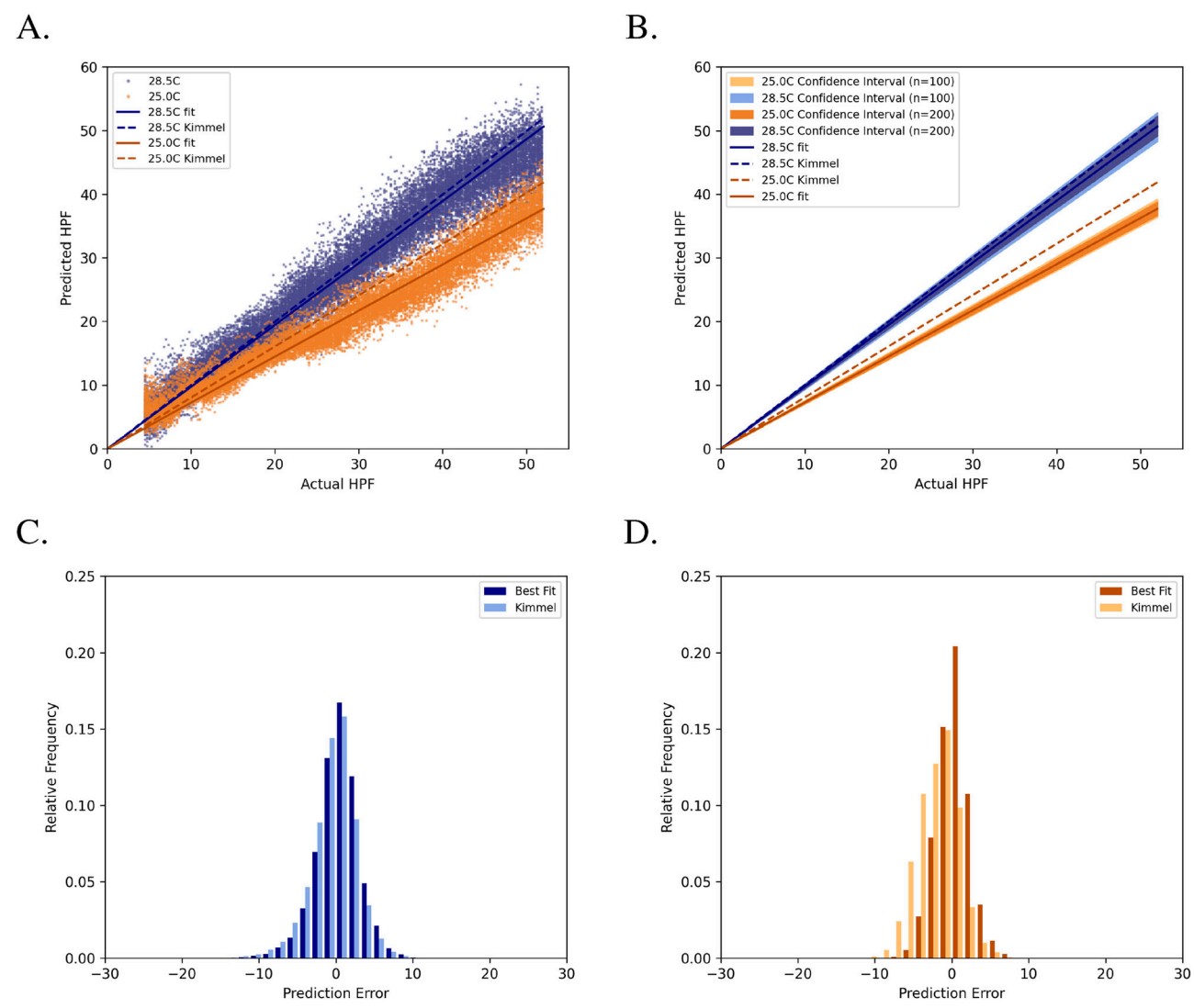

**Figure 3. KimmelNet successfully detects different developmental rates in two populations of zebrafish embryos.**
**(A)** Predictions produced by KimmelNet for Datasets "A" and "B" (Table 1) closely agree with those predicted by Equation (1). Each data point represents a single image. The solid lines represent the line of best fit (through the origin) to KimmelNet's predictions, whereas the dotted lines indicate the expected rate of development for the indicated temperatures based on Equation (1). **(B)** Confidence intervals associated with lines of best fit illustrate that, even with just a small number of images, KimmelNet can discriminate between two different populations. The inner and outer confidence intervals indicate the range of lines of best fit determined for 10,000 randomly chosen sets of 100 (outer) or 200 (inner) data points from "A." **(C, D)** Distribution of prediction errors produced by KimmelNet relative to lines of best fit (sold lines in "A") and Equation (1) (dotted lines in "A").

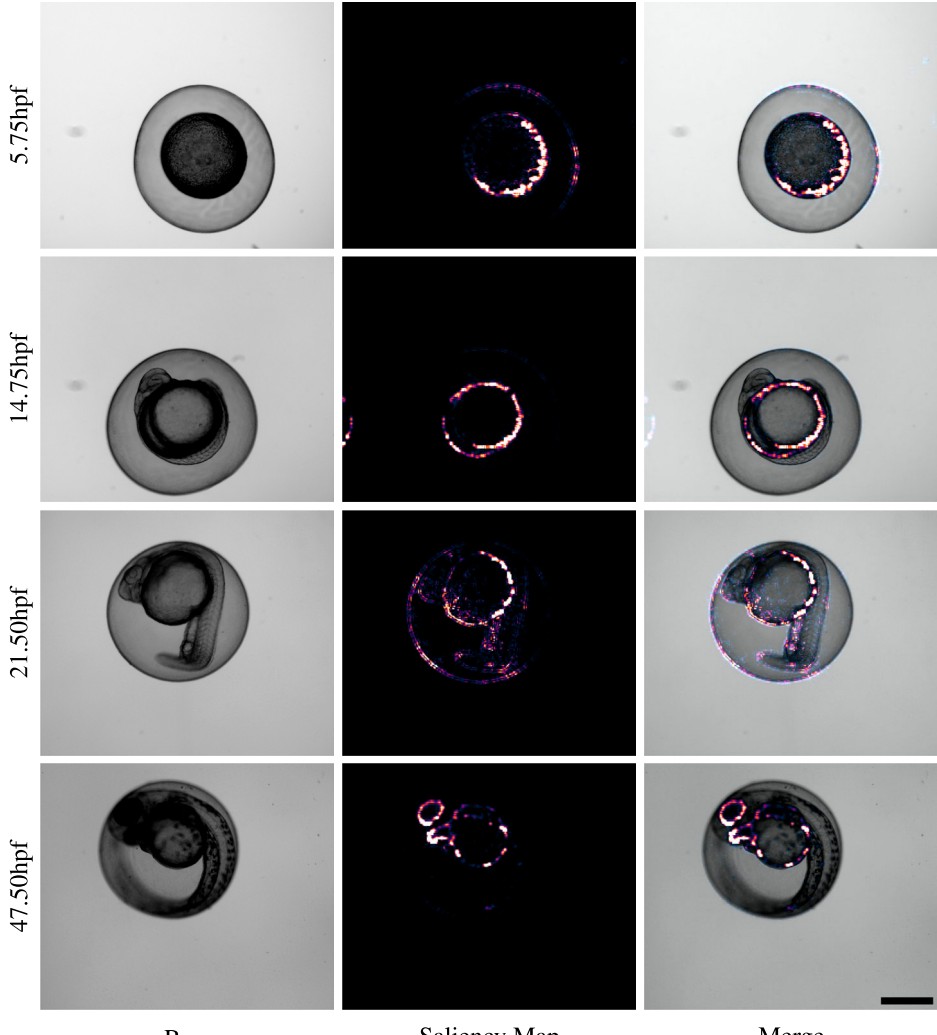

**Figure 4. Saliency maps illustrate that KimmelNet responds to image pixels relevant to the zebrafish embryos.**
Image background contributes little to predictions. Scale bar is 500 μm and all images are scaled equally.

different embryos within the same clutch can develop at different rates (Parichy et al, 2009). It is therefore very likely that images in our training data corresponding to x hpf depict embryos at slightly different developmental stages. However, the aim of this study was not to develop a tool to accurately stage individual embryos. The intention of this study was to demonstrate that, even in the absence of time-consuming manual annotation of training data, it is possible to differentiate between populations developing at different rates using deep learning.

## KimmelNet can generalise to previously unseen data

In order for a deep learning model to useful to the broader community, it needs to generalise to previously unseen data. We challenged KimmelNet with two new datasets, acquired at a different site to the training data ("C" and "D" in Table 1 and Fig 5A). Even though the predicted hpfs produced by KimmelNet contained larger errors (Fig 5D and E) than those shown in Fig 3, it was still possible to distinguish between two populations incubated at different temperatures (Fig 5C).

However, trendlines fit to the predictions in Fig 5B fit poorly ($R^2$ = 0.64 for dataset C and $R^2$ = 0.53 for dataset B) relative to Fig 3A.

## KimmelNet's predictions can be refined with transfer learning

We investigated whether the results shown in Fig 5B–E could be improved by way of transfer learning to fine-tune the model to new data. We ran a variety of different transfer learning regimes (Table 2), experimenting with the number of training images used and the proportion of KimmelNet's tuneable parameters that required updating to improve results.

We found that the number of training images used had a significant impact on the outcome of the transfer learning, whereas the proportion of parameters in the model that were updated had a comparatively small influence (Table 2 and Figs 6A–I and 7A–I). For example, using 12 wells from the original 96 well plate and retraining 23% of KimmelNet's parameters resulted in a reduction in the mean prediction error from 1.30 ± 5.52 h (Table 1 and Fig 5D) to 0.90 ± 3.85 h (Table 2 and Fig 7D).

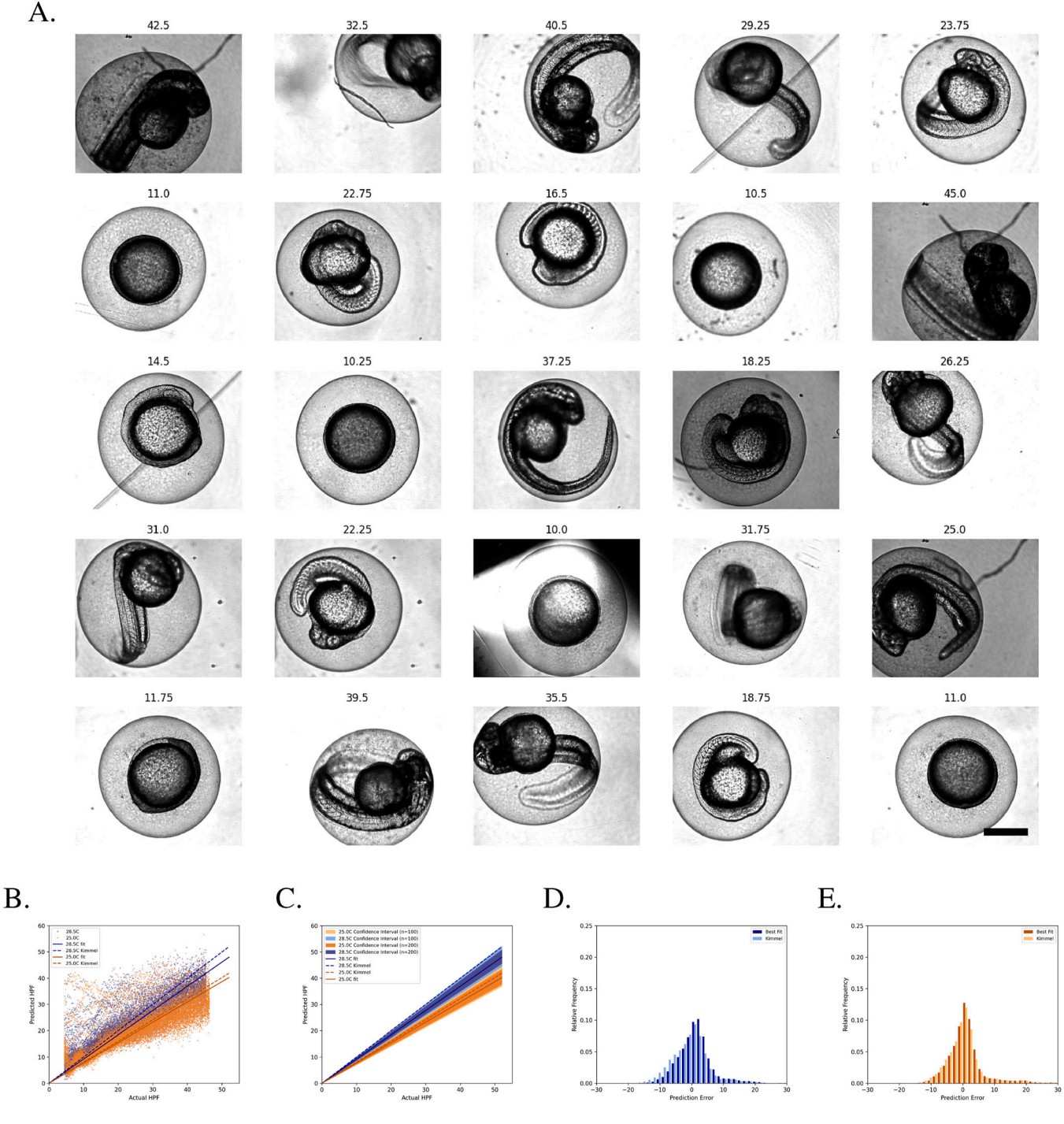

**Figure 5. KimmelNet can distinguish populations based on previously unseen data.**
**(A)** Example images of zebrafish embryos at various indicated stages of development—labels indicate hours post-fertilisation. Images were drawn from randomly selected wells in two different multi-well plates (datasets "C" and "D" in Table 1). Scale bar is 500 μm and all images are scaled equally. **(B)** Predictions produced by KimmelNet for datasets "C" and "D" (Table 1). Each data point represents a single image. The solid lines represent the line of best fit (through the origin) to KimmelNet's predictions, whereas the dotted lines indicate the expected rate of development for the indicated temperatures based on Equation (1). **(C)** Confidence intervals associated with lines of best fit illustrate that, even with just a small number of images, KimmelNet can discriminate between two different populations. The inner and outer confidence intervals indicate the range of lines of best fit determined for 10,000 randomly chosen sets of 100 (outer) or 200 (inner) data points from "A." **(D, E)** Distribution of prediction errors produced by KimmelNet relative to lines of best fit (sold lines in "A") and Equation (1) (dotted lines in "A").

**Table 2. Partitioning of dataset "C" in Table 1 for transfer learning and subsequent testing.**

| Dataset | % images used for transfer learning | % model parameters retrained | Test prediction errors (mean ± sd) | Slope of line fit to test predictions | $R^2$ of line fit to test predictions |
|---|---|---|---|---|---|
| A* | 38.5 | 23 | 0.64 ± 4.16 | 0.938 | 0.847 |
| B* | 38.5 | 41 | 0.60 ± 3.83 | 0.955 | 0.874 |
| C* | 38.5 | 68 | 0.48 ± 3.84 | 0.957 | 0.881 |
| D* | 12.5 | 23 | 0.90 ± 3.85 | 0.93 | 0.843 |
| E* | 12.5 | 41 | 0.72 ± 3.69 | 0.925 | 0.866 |
| F* | 12.5 | 68 | 0.65 ± 3.85 | 0.932 | 0.864 |
| G* | 2 | 23 | 1.47 ± 5.33 | 1.007 | 0.7 |
| H* | 2 | 41 | 1.42 ± 5.42 | 1.004 | 0.702 |
| I* | 2 | 68 | 1.43 ± 5.53 | 1.015 | 0.699 |

Overall, these results demonstrate that KimmelNet can be used "as-is" without any further training on new data. However, in the event that subtle differences between populations cannot be elucidated, a minimal amount of transfer learning can significantly improve the predictive power of the model.

In conclusion, the developing zebrafish embryo is used in many different types of studies and accurate staging is essential. When comparing an experimental group of embryos with a control group, ensuring the embryos have reached the same developmental stage allows for meaningful comparisons to be made. KimmelNet enables rapid, unbiased assessment of thousands of images with minimal time commitment. We anticipate that our classifier will be a useful tool for the zebrafish community. There is also potential for KimmelNet to be useful in other developmental contexts, by retraining with images of other organisms.

# Materials and Methods

### Image data acquisition

All images used for training and testing are summarised in Table 1. The details of acquisition for the images comprising datasets "A" and "B" can be found in the study of Jones et al (2022). Datasets "C" and 'D" were acquired as follows.

#### *Zebrafish husbandry*
Zebrafish were maintained according to the protocols outlined by Princeton University Institutional Animal Care and Use Committee (Approval number: 1915). Wild-type embryos were collected from the PWT strain generated in Burdine laboratory (Schottenfeld et al, 2007). Male and female adult fish separated by barriers were set up for breeding the night before the experiment. In the morning upon lights on, embryos laid within 30 min of opening the barriers were collected and treated as a single clutch. Embryos were raised at 28°C in 1x E3 embryo medium without methylene blue until plating for microscopy.

#### *Live imaging*
Zebrafish embryos were maintained at 28°C until 2–3 hpf when they were transferred into a U-bottomed 96-well plate in E3 medium.

The plate was covered with an FEP membrane (1 mil Teflon FEP film, American Durafilm) to prevent condensation and allow for gas exchange. Brightfield images of embryos individually seeded in the 96-well plate were acquired every 15 min starting at 2.5–3.5 hpf for ~43 h on a Nikon Ti-E inverted microscope with a Hamamatsu BT-Fusion sCMOS camera, using a 4X/0.13 Plan Fluor objective. Sample temperature was maintained at either 25.0 or 28.5°C using a Tokaii Hit stage top incubator. The microscope was controlled with Nikon NIS-Elements Software. Images were manually checked after capture to ensure embryo health.

### Data partitioning and augmentation

Dataset "A" (Table 1) was split randomly into training and test data on a well-by-well basis. Certain wells were manually excluded if it was apparent that the embryos died or were significantly out of view—the complete list of excluded wells is explicitly listed in the code (see https://github.com/djpbarry/KimmelNET/blob/main/train_model.py).

Six data augmentation layers were used for training purposes to prevent over-fitting to the training data (Fig 1). These consisted of randomly applied histogram equalisation, randomly applied image saturation, randomly added Gaussian noise, random horizontal and vertical image flipping, random translation in the x-direction (max 20% of image width), and random zoom (maximum 30%). Random translation in the y-direction was avoided because of the aspect ratio of the images (width > height)—due to the centre-crop layer only cropping in x (but not y), any artifacts introduced by translation in the x-direction would be largely removed, but those introduced by translation in y would not.

### Network training

KimmelNet was trained on a single GPU node on the Francis Crick Institute's HPC platform (Crick data Analysis and Management Platform). Each node consists of 4 NVIDIA Tesla V100 SXM2 graphics cards. KimmelNet was trained on all four graphics cards simultaneously using Tensorflow's distribute library. Training was performed using the Adam optimiser (Kingma & Ba, 2014 Preprint) with a learning rate of 0.0005 and mean squared error as the loss function.

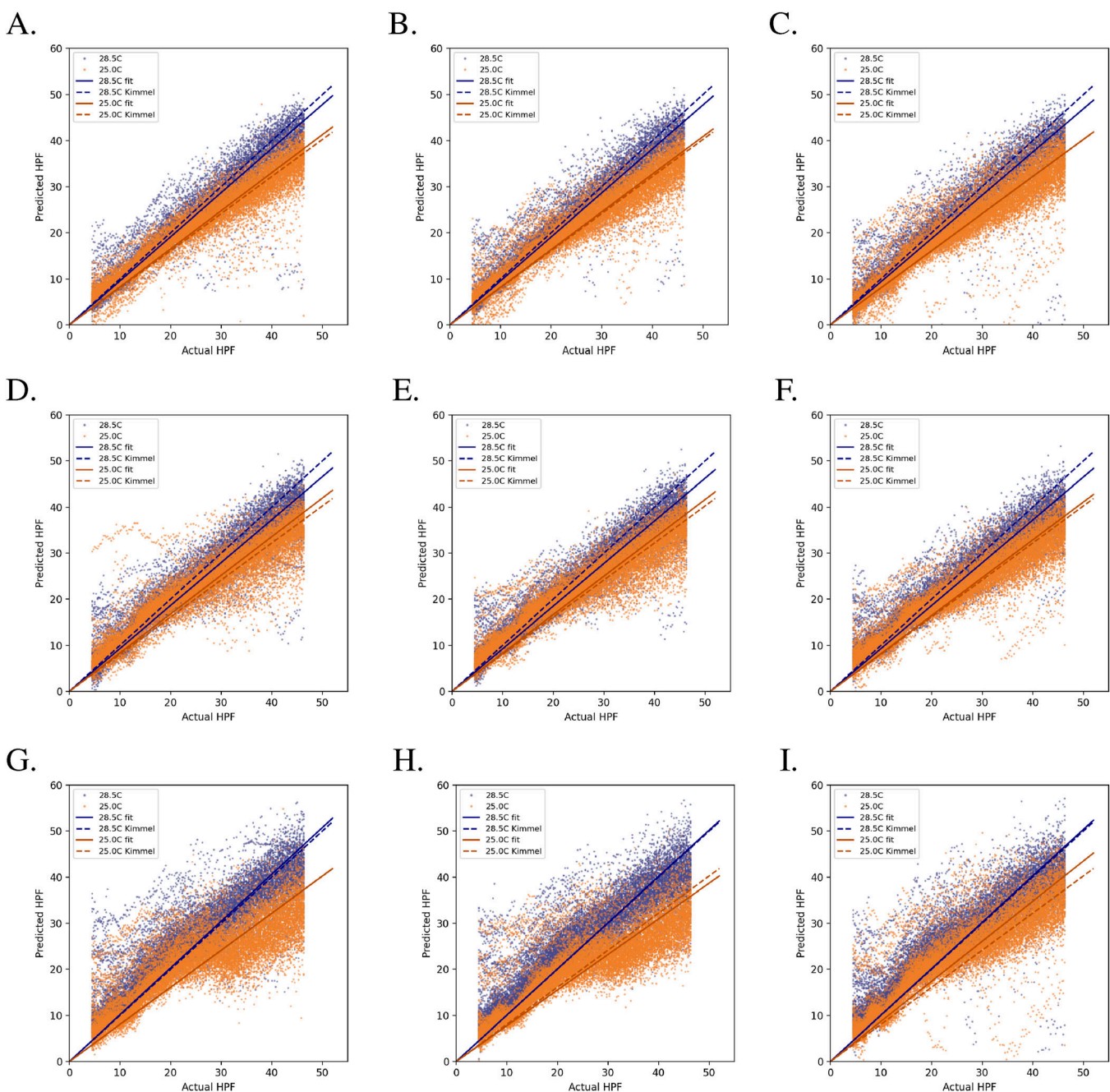

**Figure 6. Transfer learning results in more linear predictions by KimmelNet.**
Labels (A, B, C, D, E, F, G, H, I) correspond to datasets in Table 2. The number of images used for transfer learning increases when moving from bottom to top. The number of network parameters retrained increases when moving from left to right.

### Bootstrapping test predictions

To test the correspondence between predictions made by Kimmelnet and those made by the equation published by Kimmel et al (1995), test predictions were fit to a straight line through the origin of the form $y = mx$. To calculate confidence intervals associated with these fits, a series of 10,000 line fits were performed, each with a different random sample of the test data, consisting of either 100 or 200 data points. The maximum and minimum values of $m$ produced by these line fits were then interpreted as a confidence range for the line fit to the complete data set.

### Transfer learning

Transfer learning was performed on the Francis Crick Institute's HPC platform, using an individual graphics card. Training was

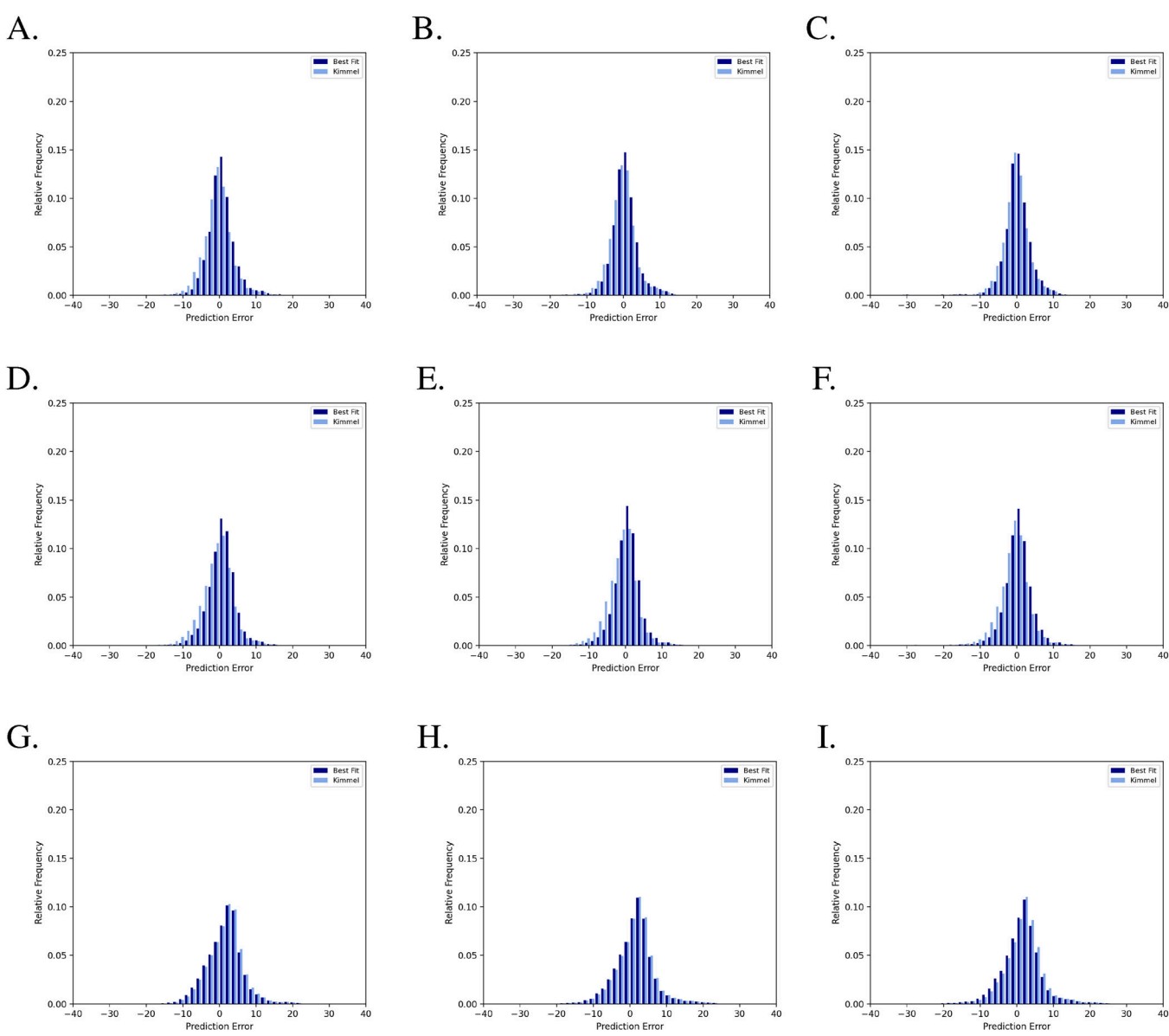

**Figure 7. Transfer learning reduces the magnitude of errors produced by KimmelNet.**
Labels (A, B, C, D, E, F, G, H, I) correspond to datasets in Table 2. The number of images used for transfer learning increases when moving from bottom to top. The number of network parameters retrained increases when moving from left to right.

performed for 1,200 epochs using the Adam optimiser (Kingma & Ba, 2014 *Preprint*) with a learning rate of 0.0001 and mean squared error as the loss function. Note that the learning rate is five times lower than that used for the initial training, to prevent the model converging too quickly to a local minimum. Data were partitioned into training and test datasets (Table 2) by randomly selecting wells for training.

## Statistical analysis

Calculation of mean and SD was performed using the mean and std functions in NumPy (Harris et al, 2020). The SEM was calculated using the sem function in the SciPy stats module (Virtanen et al, 2020).

## Code implementation

KimmelNet was implemented using Python 3.9 (www.python.org) and TensorFlow 2.9 (www.tensorflow.org). All plots in this manuscript were produced using Matplotlib 3.5 (Hunter, 2007) and all curve-fitting performed with SciPy 1.8 (Virtanen et al, 2020). Training and testing were performed with NVIDIA CUDA 11.2 (https://developer.nvidia.com/cuda-zone) and cuDNN 8.1 (https://developer.nvidia.com/cudnn).

## Software availability

All code for building, training, and testing KimmelNet is publicly available online (https://github.com/djpbarry/KimmelNET). A Binder implementation is also available to test KimmelNet on a subset of our test data (https://mybinder.org/v2/gh/djpbarry/KimmelNET/main?labpath=zebrafish_age_estimator.ipynb).

# Data Availability

All image data used in this study are available to download from the BioImage Archive (accession numbers S-BIAD531 and S-BIAD840).

# Supplementary Information

# Acknowledgements

We are grateful for the support provided by the Princeton Core Imaging Facility, particularly Gary S Laevsky, and Triveni Menon of Rebecca D Burdine's laboratory in Princeton for help with provision of zebrafish embryos. We would also like to thank Danelle Devenport for the provision of additional funding. This work was supported by the Francis Crick Institute, which receives its core funding from Cancer Research UK (CC1157, CC0199), the UK Medical Research Council (CC1157, CC0199), and the Wellcome Trust (CC1157, CC0199).

## Author Contributions

RA Jones: conceptualization, validation, investigation, methodology, and writing—review and editing.
MJ Renshaw: resources, validation, investigation, and methodology.
DJ Barry: conceptualization, data curation, software, formal analysis, validation, investigation, visualization, methodology, and writing—original draft.

## Conflict of Interest Statement

The authors declare that they have no conflict of interest.

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
