## [Reviewer comments · Life Science Alliance]

Life Science Alliance

Automated staging of zebrafish embryos with deep learning

Rebecca Jones, Matthew Renshaw, and David Barry

DOI: <https://doi.org/10.26508/lsa.202302351>

Corresponding author(s): David Barry, The Francis Crick Institute

Review Timeline:

Submission Date:	2023-09-01
Editorial Decision:	2023-10-03
Revision Received:	2023-10-14
Accepted:	2023-10-18

Transaction Report:

Please note that the manuscript was reviewed at Review Commons and these reports were taken into account in the decision-making process at *Life Science Alliance*.

Review
COMMONS

Reviews

Review #1

In this manuscript the author is presenting a deep-learning model used to predict the development stage of zebrafish embryo. A robust method that can accurately classify a zebrafish into different development stages is highly relevant for many researchers working with zebrafish and hence the importance in developing methods like this is high.

The manuscript is overall ok. However, more data is needed to convince the reader that the method is robust enough to work with other samples in other labs. This would greatly improve the impact of the publication.

Page 6.

- How is the data acquired?

Page 8.

"This indicates that while KimmelNet can be used successfully with noisier test data than that on which it was trained, there is an upper limit to how noisy the data can be."

- This is an obvious statement there will always be an upper limit on noise.

Page 9.

- Are only wildtype embryos used? How would this work on different mutants. To evaluate the robustness of the method this it would be valuable to test on some mutant line with known developmental difference from the wild type.

Image data.

- I would strongly suggest that the author should include a description of the data in the manuscript. A description of how the data is acquired, microscope, different batches, type of embryos used.

"Random 160 translation in the y-direction was avoided due to the aspect ratio of the images (width > 161 height) - any artifacts introduced by translation in the x-direction would be removed by the 162 centre crop layer, but this would not be the case for translation in the y-direction."

- Could this be solved by using some border method reflection, repetition or fixed value?

Page 10.

Addition of Noise to Image Data

- This should be added in the training phase. This would probably improve the robustness of the network and also improve the results on the test data.

- Supplementary 3 images with high noise. It is worrying that the network is not able to handle the noise in this figure. Looks like the features that is used to distinguish the developmental stage of the embryo is still clearly seen with this high noise level? Retrain the model with noise as an augmentation to improve this.

The development of methods like this is highly relevant in the zebrafish community. Staging and evaluating the developmental stage for zebrafish is common and is of interest to the broad community. A lot of this work today is done manually, limiting the throughput, and adding human bias.

The limit of this study is the dataset used for training and evaluation. Firstly, it is not clear about the structure of the data and how it is acquired, different types of fish or imaging setup etc. For a method to be useful to the community it needs to be robust enough to handle different types of fish (transgenic lines). The manuscript would be greatly improved by adding this to the training and evaluation.

Review #2

****Summary****

The paper "Automated staging of zebrafish embryos with KimmelNet" by Barry et al., presents a method to automatically stage developmental timepoints of zebrafish embryos based on convolutional neural networks (CNN). The authors show that a CNN trained on ~20k images can determine time post fertilization on test-image sets with

an accuracy on the range of a few hours. This technique undoubtedly has the potential to become very useful for any zebrafish researchers interested in developmental timing as it eases analysis and removes potential subjective bias.

****Major comments****

In its current form the paper lacks sufficient graph annotations and method descriptions. This makes it hard in places to judge the validity of the claims. I do believe that the presented method can be useful and is likely valid but to be convincing, the authors need to spend more time expanding the methods, justifying their choices of analysis and clarifying figure annotations.

****Specific points:****

1. The annotation of the training data is not described and specifically it is unclear how valid the staging of the training data itself is. The authors state in the introduction "the hours post fertilization (hpf) [...] provides only an approximation of the actual developmental stage". It is therefore critical to know how this was accounted for in the annotation of the training data. Since the quality of the training data will ultimately limit the best-case quality of Kimmel Net. The authors need to go into some detail here even though the training data appears to be from another published dataset.
2. Why were "test predictions fit to a straight line through the origin". On the one hand this makes sense (since the slope would indicate the correspondence) but it should be clarified why an intercept was omitted in the fit. After all it is unclear if Kimmel net correctly identifies 0Hpf embryos.
3. The methods do not list how the mean of the absolute error was calculated from 3B/C. I think this should be the mean of the absolute error (not the mean of the error) but in that case the numbers listed in the text appear rather small given the histograms in 3 B/C. A clear statement in the methods would clarify this issue.

****Minor comments****

1. The Y-axis in Figure 2B is simply labeled "Loss" - what is the unit of this loss? HPF? Or HPF^2 ? This is important for judging the quality of the fit
2. Figure 3 B. I would suggest changing the labels of the confidence intervals in the legend. "Inner and outer" is already clear from the figure itself, so labels that state that these are derived from $n=100$ vs. $n=20$ test image sized samples would be more useful to the reader

****Referees cross-commenting****

I concur with comments issued by the other reviewers. I think it will be especially important to address the comments related to testing the method on mutants (Reviewer #1) and training the model in the presence of noise to increase its robustness (Reviewers #1 and #3) as well as addressing the overall annotation/generation of the training data (Reviewers #1 and #2).

I think these points will be critical to make the paper useful by increasing transparency and ensuring reproducibility in other labs with different imaging conditions, strains, mutants, etc.

Developmental delay is a common occurrence that can be caused by genetic and environmental background effects. It is therefore highly desirable to properly quantify this variable. The work presented here makes an important step in this direction, by allowing to quantify developmental timepoints independent of subjective staging. This speeds up analysis, increases reproducibility and enhances rigor. However, as my comments above indicate, the significance also depends on the ability of potential users to judge the quality of the work. Once those issues have been addressed, I think the work will be of broad interest to the developmental biology community, first and foremost labs utilizing the zebrafish model. However, as the authors state, the presented model architecture could be trained with the data from other species as well.

Review #3

****Summary:****

Properly staging embryos of zebrafish embryos is important, yet provides challenging since it can depend on many

factors, such as temperature, water quality, fish population density, etc. Here, the authors provide a deep-learning-based model called KimmelNet that allows the prediction of the age of zebrafish embryos, using 2D brightfield images. The technique is robust to weak measurement noise and can also be used to identify developmental delays from a very small number of experimental data.

The code is accessible to the reader, open-source and should be useable by experimentalists without huge effort.

****Major comments:****

I suggest retraining the model and application of the model to additional data for the following reasons:

- Why did the authors not train for (high) measurement noise and heterogeneous background illumination? Would that not make the model more robust? In principle, creating training should not be considerably harder than before, since the manipulation of the images has been already shown in the manuscript and the authors just need to run it again on the HPC cluster. If there are no technical or administrative constraints (access to the cluster, computational effort, high costs, etc.), the authors should retrain their model.
- For Fig. S2 and S3 it is not clear if there is such a strong deviation from the Kimmel equation due to measurement noise or due to the background illumination. The saliency maps appear as if they are mainly affected by the illumination, and maybe less by the noise. Would it be possible to apply the model to a case without artificial noise, but with heterogeneous background illumination to identify what has a bigger impact?

Additionally, the authors need to clarify what exactly they are comparing in this manuscript and rework their interpretation of their findings:

- When comparing the predictions between KimmelNet and the Kimmel equation, the authors use an equation of the form $y=mx$. Could the authors please elaborate on why they introduce the constraint of $y(0)=0$? It might be naturally given by the so-called Kimmel equation, but by looking at Fig 3a, it seems like an equation of the form $y=mx+a$ would be more appropriate and it appears like KimmelNet introduces an offset of around $a=2h$ for 25 Celsius. The authors need to discuss this.
- In lines 5-8 the authors say that developmental stage progression does not only depend on temperature, but also on population density, water quality etc. and they explain that usually not only hpf, but also staging guides based on morphological criteria are used! If that is true, how good is their training data set that only uses hpf and not the other important guides? How did the authors test that these factors have no impact on their training data?

Since this tool has the potential to have a big impact on zebrafish research, it would be nice to provide some examples of how exactly this could be achieved:

- Could the authors discuss how exactly their tool is useful to experimentalists? Is it the idea that if an experimentalist wants to investigate an embryo of a particular stage, they apply KimmelNet to images of embryos to identify the stage of the embryo and only then undertake their planned experiment? Is that a realistic undertaking?
- Would it be possible to provide a tutorial (or even video tutorial if such skills are available in the group of authors) that provides real examples of how to apply the technique? This would make it easier for people without advanced Python/Deep-Learning skills to use the tool, hence improving the impact of KimmelNet.

I am very critical towards equation 1. Please note that I don't think this has any impact on the quality of the technique provided in this manuscript and the significant flaws can already be found in Kimmel 1995 (which is not under review here). That is why I suggest rewriting of this manuscript to not support an over-interpretation of this equation.

- I do not think that the Kimmel equation is an established term. At least a Google Scholar Search for "Kimmel equation" only gives one result: the preprint of this manuscript.
- The equation has no mathematical meaning regarding its units (subtracting temperature and a unitless value). I also very rarely see equations with Degrees Celsius and not Kelvin.
- Additionally, the equation provides a linear relationship between the development time and temperature $h(T)$ and in Kimmel et al, it is shown that this is only true for 25-33 Celsius. Such a linearisation is not very surprising for a small temperature range, but I am not sure how true it is for higher temperature differences. Hence, I feel that it is very bold to give a specific name to such an equation, giving it an importance that it does not deserve.

****Minor comments:****

- For the measurement noise cases it would be nice to have some example images of fish with the specific noise levels in Fig S1 and Fig S2.
- Could the authors hypothesize why they predict a slower dynamic for 25 Celsius than predicted by the Kimmel

equation?

Strengths of the study:

An easy-to-use method to automatically stage zebrafish embryos and identify differences in the developmental stage is very important for the zebrafish community and the technique in this manuscript definitely novel. The tool is can be used without large effort and the authors suggest that it can also find applications beyond zebrafish embryos. Hence, it is not only interesting to the zebrafish community, but to a broader developmental biology audience.

Weakness of the study:

The main weakness of the manuscript is in the training data used for the deep-learning model and the apparent large impact of heterogeneous background illumination. If that is not solved, it is unclear if this technique will find an application in the zebrafish community.

Reviewer #1 (Evidence, reproducibility and clarity (Required)):

In this manuscript the author is presenting a deep-learning model used to predict the development stage of zebrafish embryo. A robust method that can accurately classify a zebrafish into different development stages is highly relevant for many researchers working with zebrafish and hence the importance in developing methods like this is high.

The manuscript is overall ok. However, more data is needed to convince the reader that the method is robust enough to work with other samples in other labs. This would greatly improve the impact of the publication.

We agree with the reviewer and have included in our revised manuscripts additional test data that was acquired at a different laboratory to the training data (Figures 5 - 7).

Page 6.

- How is the data acquired?

Images used to do initial model training are the same as those used in a previous study - the details of image acquisition are contained in the relevant publication (doi: 10.12688/wellcomeopenres.18313.1). However, we have now added "Zebrafish Husbandry" and "Live Imaging" for newly-acquired images. We have added a table (Table 1) listing details of all image data used in the study.

Page 8.

"This indicates that while KimmelNet can be used successfully with noisier test data than that on which it was trained, there is an upper limit to how noisy the data can be."

- This is an obvious statement there will always be an upper limit on noise.

We agree with the reviewer that this statement is not terribly informative. This section ("KimmelNet's prediction accuracy is not significantly impacted by moderate levels of additive noise") has been removed from the revised manuscript in favour of incorporating a section detailing testing of the model on newly-acquired images ("KimmelNet can generalise to previously unseen data").

Page 9.

- Are only wildtype embryos used? How would this work on different mutants. To evaluate the robustness of the method this it would be valuable to test on some mutant line with known developmental difference from the wild type.

We agree with the reviewer that testing on a mutant line would lend more weight to our findings. For example, the p53^{-/-} zebrafish has a reported, published developmental delay, but we did not have access to that line. However, the developmental delay reported for the p53^{-/-} mutant is virtually

indistinguishable from that effected by a temperature change. We therefore focussed our efforts on developmental delay affected by altering incubation temperature only.

Image data.

- I would strongly suggest that the author should include a description of the data in the manuscript. A description of how the data is acquired, microscope, different batches, type of embryos used.

The image data used in the first draft of the manuscript is the same as that used in a previous publication (Jones et al. 2022), which contains all the relevant details the reviewer seeks. However, we have now added the relevant information for the newly-acquired image data.

"Random160translation in the y-direction was avoided due to the aspect ratio of the images (width>161height) - any artifacts introduced by translation in the x-direction would be removed by the162centre crop layer, but this would not be the case for translation in the y-direction."

- Could this be solved by using some border method reflection, repetition or fixed value?

The reviewer is correct that some form of image reflection or repetition could be utilised. However, given the nature of our images, if an embryo is located close to the image boundary, reflection/repetition can result in some odd artefacts, so we minimised the use of such fill methods (also used by the random zoom augmentation layer). We could instead use an arbitrary fixed value, as the reviewer suggested, but finding a value suitable for all images is difficult.

Page 10.

Addition of Noise to Image Data

- This should be added in the training phase. This would probably improve the robustness of the network and also improve the results on the test data.

We agree with the reviewer and have now added a random Gaussian noise layer for data augmentation purposes during model training (see Figure 1).

- Supplementary 3 images with high noise. It is worrying that the network is not able to handle the noise in this figure. Looks like the features that is used to distinguish the developmental stage of the embryo is still clearly seen with this high noise level? Retrain the model with noise as an augmentation to improve this.

As the reviewer suggested, addition of random noise is now incorporated into model training. The new version of the manuscript does not include the same supplemental figures, but instead includes additional testing on newly-acquired data instead.

Reviewer #1 (Significance (Required)):

The development of methods like this is highly relevant in the zebrafish community. Staging and evaluating the developmental stage for zebrafish is common and is of interest to the broad community. A lot of this work today is done manually, limiting the throughput, and adding human bias.

The limit of this study is the dataset used for training and evaluation. Firstly, it is not clear about the structure of the data and how it is acquired, different types of fish or imaging setup etc. For a method to be useful to the community it needs to be robust enough to handle different types of fish (transgenic lines). The manuscript would be greatly improved by adding this to the training and evaluation.

We have now added additional datasets for the purposes of evaluating the model.

Reviewer #2 (Evidence, reproducibility and clarity (Required)):

Summary

The paper "Automated staging of zebrafish embryos with KimmelNet" by Barry et al., presents a method to automatically stage developmental timepoints of zebrafish embryos based on convolutional neural networks (CNN). The authors show that a CNN trained on ~20k images can determine time post fertilization on test-image sets with an accuracy on the range of a few hours. This technique undoubtedly has the potential to become very useful for any zebrafish researchers interested in developmental timing as it eases analysis and removes potential subjective bias.

Major comments

In its current form the paper lacks sufficient graph annotations and method descriptions. This makes it hard in places to judge the validity of the claims. I do believe that the presented method can be useful and is likely valid but to be convincing, the authors need to spend more time expanding the methods, justifying their choices of analysis and clarifying figure annotations.

We believe that we have addressed the reviewer's concerns in this revised manuscript, as detailed in response to the specific points below.

Specific points:

1) The annotation of the training data is not described and specifically it is unclear how valid the staging of the training data itself is. The authors state in the introduction "the hours post fertilization (hpf) [...] provides only an approximation of the actual developmental stage". It is therefore critical to know how this was accounted for in the annotation of the training data. Since the quality of the training data will ultimately limit the best-case quality of Kimmel Net. The authors need to go into some detail here even though the training data appears to be from another published dataset.

The reviewer raises a valid point – two individual zebrafish embryos that are x hours post-fertilisation are not necessarily at the same developmental stage. However, we believe it is reasonable to assume that

two *populations* of embryos x hours post-fertilisation are, on average, at the same developmental stage and it is this assumption that forms the basis for our approach. Given the inherent variability the reviewer refers to, we are not suggesting that our model would be particularly accurate for staging individual embryos. However, we are very confident (and we believe the data in the manuscript supports this) that given a population of embryos, our model will provide an accurate rate of development. We have added a paragraph (lines 131-141) to address this point.

2) Why were "test predictions fit to a straight line through the origin". On the one hand this makes sense (since the slope would indicate the correspondence) but it should be clarified why an intercept was omitted in the fit. After all it is unclear if Kimmel net correctly identifies 0Hpf embryos.

The reviewer makes a valid point – we do not know what predictions KimmelNet would produce for images of embryos closer to 0 hpf. However, we felt an equation of the form $y=mx$ was a reasonable choice for two reasons. First of all, it matches the form of the Kimmel equation, which, despite its flaws, we are using as a benchmark of sorts – the absence of a y intercept makes comparisons with the Kimmel equation straightforward. Secondly, a “perfect” model would produce a straight line fit with $y=x$, so the lack of a y intercept seemed a reasonable constraint to impose. We have added some brief text (lines 103-105) to clarify this choice.

3) The methods do not list how the mean of the absolute error was calculated from 3B/C. I think this should be the mean of the absolute error (not the mean of the error) but in that case the numbers listed in the text appear rather small given the histograms in 3 B/C. A clear statement in the methods would clarify this issue.

We have now added a “Statistical Analysis” section under Materials & Methods to detail exactly what was used to calculate the values given for error analysis. We have calculated the mean of the error, not the mean of the absolute error, as we wish to illustrate that the mean is close to zero. We have used the standard deviation of the errors to illustrate that there is a significant spread in the error values, as depicted in Figure 3C and D.

Minor comments

1) The Y-axis in Figure 2B is simply labeled "Loss" - what is the unit of this loss? HPF? Or HPF²? This is important for judging the quality of the fit

We thank the reviewer for drawing our attention to this omission. The loss is hpf^2 (mean squared error) and we have updated the plot to reflect this.

2) Figure 3 B. I would suggest changing the labels of the confidence intervals in the legend. "Inner and outer" is already clear from the figure itself, so labels that state that these are derived from n=100 vs. n=20 test image sized samples would be more useful to the reader

We thank the reviewer for this suggestion – we have updated the figure legend accordingly.

****Referees cross-commenting****

I concur with comments issued by the other reviewers. I think it will be especially important to address the comments related to testing the method on mutants (Reviewer #1) and training the model in the presence of noise to increase its robustness (Reviewers #1 and #3) as well as addressing the overall annotation/generation of the training data (Reviewers #1 and #2).

We believe we have now addressed all of these concerns. The model has been retrained with additional data augmentation incorporating random noise, tested on newly-acquired data and we have added tables summarising the details of all image data used in this study.

I think these points will be critical to make the paper useful by increasing transparency and ensuring reproducibility in other labs with different imaging conditions, strains, mutants, etc.

Reviewer #2 (Significance (Required)):

Developmental delay is a common occurrence that can be caused by genetic and environmental background effects. It is therefore highly desirable to properly quantify this variable. The work presented here makes an important step in this direction, by allowing to quantify developmental timepoints independent of subjective staging. This speeds up analysis, increases reproducibility and enhances rigor. However, as my comments above indicate, the significance also depends on the ability of potential users to judge the quality of the work. Once those issues have been addressed, I think the work will be of broad interest to the developmental biology community, first and foremost labs utilizing the zebrafish model. However, as the authors state, the presented model architecture could be trained with the data from other species as well.

We thank the reviewer for their positive feedback.

Reviewer #3 (Evidence, reproducibility and clarity (Required)):

Summary:

Properly staging embryos of zebrafish embryos is important, yet provides challenging since it can depend on many factors, such as temperature, water quality, fish population density, etc. Here, the authors provide a deep-learning-based model called KimmelNet that allows the prediction of the age of zebrafish embryos, using 2D brightfield images. The technique is robust to weak measurement noise and can also be used to identify developmental delays from a very small number of experimental data.

The code is accessible to the reader, open-source and should be useable by experimentalists without huge effort.

Major comments:

I suggest retraining the model and application of the model to additional data for the following reasons:

- Why did the authors not train for (high) measurement noise and heterogeneous background illumination? Would that not make the model more robust? In principle, creating training should not be considerably harder than before, since the manipulation of the images has been already shown in the manuscript and the authors just need to run it again on the HPC cluster. If there are no technical or administrative constraints (access to the cluster, computational effort, high costs, etc.), the authors should retrain their model.*

We thank the reviewer for this suggestion. As detailed in Figure 1, with a view to making the model more robust, we have now added several more layers of data augmentation, including the addition of random noise, and retrained our model.

- For Fig. S2 and S3 it is not clear if there is such a strong deviation from the Kimmel equation due to measurement noise or due to the background illumination. The saliency maps appear as if they are mainly affected by the illumination, and maybe less by the noise. Would it be possible to apply the model to a case without artificial noise, but with heterogeneous background illumination to identify what has a bigger impact?*

We thank the reviewer for this suggestion. We have now replaced the “artificial” examples used in the previous version of the manuscript with newly-acquired data (Figure 5), which exhibits different characteristics to that used for training.

Additionally, the authors need to clarify what exactly they are comparing in this manuscript and rework their interpretation of their findings:

- When comparing the predictions between KimmelNet and the Kimmel equation, the authors use an equation of the form $y=mx$. Could the authors please elaborate on why they introduce the constraint of $y(0)=0$? It might be naturally given by the so-called Kimmel equation, but by looking at Fig 3a, it seems like an equation of the form $y=mx+a$ would be more appropriate and it appears like KimmelNet introduces an offset of around $a=2h$ for 25 Celsius. The authors need to discuss this.*

The main rationale for using an equation of the form $y=mx$ is to be consistent with the Kimmel equation (see lines 103-105). The reviewer is correct that an equation of the form $y=mx+c$ may well produce a better fit to the data, but omitting a y intercept makes comparison with the Kimmel Equation trivial.

- In lines 5-8 the authors say that developmental stage progression does not only depend on temperature, but also on population density, water quality etc. and they explain that usually not only hpf, but also staging guides based on morphological criteria are used! If that is true, how good is their*

training data set that only uses hpf and not the other important guides? How did the authors test that these factors have no impact on their training data?

We have now added a paragraph (lines 131-141) to address this point.

Since this tool has the potential to have a big impact on zebrafish research, it would be nice to provide some examples of how exactly this could be achieved:

- *Could the authors discuss how exactly their tool is useful to experimentalists? Is it the idea that if an experimentalist wants to investigate an embryo of a particular stage, they apply KimmelNet to images of embryos to identify the stage of the embryo and only then undertake their planned experiment? Is that a realistic undertaking?*

As is evidenced by the errors presented in Figure 3C & D, testing KimmelNet on individual images of embryos may well result in a large error in the predicted hpf. As such, it is not appropriate to use the tool in such a manner. However, to modify the example provided by the reviewer, should an experimentalist have a *population* of embryos they wished to stage, then yes, KimmelNet would certainly be an appropriate tool for doing so.

- *Would it be possible to provide a tutorial (or even video tutorial if such skills are available in the group of authors) that provides real examples of how to apply the technique? This would make it easier for people without advanced Python/Deep-Learning skills to use the tool, hence improving the impact of KimmelNet.*

A lack of user-friendly interfaces for applying deep learning methods in biology is well-documented – basic knowledge of python and tools like jupyter notebooks are often necessary (<https://doi.org/10.1038/s41592-023-01900-4>). However, we have endeavoured to make the running of KimmelNet as easy as possible for new users. A jupyter notebook instance can be run on Binder with absolutely no set-up required. To run KimmelNet on their own data, biologists just need to download the Git repo and replace the test images with their own data. We have updated the landing page on the GitHub repo to provide more specific step-by-step instructions for each of these tasks. We will also endeavour to upload our model to the BioImage Model Zoo (<https://bioimage.io/#/>) to further increase accessibility.

I am very critical towards equation 1. Please note that I don't think this has any impact on the quality of the technique provided in this manuscript and the significant flaws can already be found in Kimmel 1995 (which is not under review here). That is why I suggest rewriting of this manuscript to not support an over-interpretation of this equation.

- *I do not think that the Kimmel equation is an established term. At least a Google Scholar Search for "Kimmel equation" only gives one result: the preprint of this manuscript.*

- *The equation has no mathematical meaning regarding its units (subtracting temperature and a unitless value). I also very rarely see equations with Degrees Celsius and not Kelvin.*

- *Additionally, the equation provides a linear relationship between the development time and temperature $h(T)$ and in Kimmel et al, it is shown that this is only true for 25-33 Celsius. Such a linearisation is not very surprising for a small temperature range, but I am not sure how true it is for higher temperature differences. Hence, I feel that it is very bold to give a specific name to such an equation, giving it an importance that it does not deserve.*

We appreciate the reviewer's concerns and have removed explicit references to "The Kimmel Equation", without substantively changing the content of the manuscript.

Minor comments:

- *For the measurement noise cases it would be nice to have some example images of fish with the specific noise levels in Fig S1 and Fig S2.*

We have now removed the "synthetic" additive noise test data, previously depicted in Figures S1-3, in favour of newly-acquired images in Figures 5-7.

- *Could the authors hypothesize why they predict a slower dynamic for 25 Celsius than predicted by the Kimmel equation?*

Referring to Figure 2 in Kimmel et al (1995), it is apparent that the straight lines are by no means perfect fits to the datapoints. In Fig 2A in particular, some datapoints for the 25C data fall well below the line fit. While the published equation suggests a rate of development 80.5% of the rate at 28.5C, according to Fig 2A, an alternative line fit could give a developmental rate as low as 70-75%, which would be in agreement with our data.

Reviewer #3 (Significance (Required)):

Strengths of the study:

An easy-to-use method to automatically stage zebrafish embryos and identify differences in the developmental stage is very important for the zebrafish community and the technique in this manuscript definitely novel. The tool is can be used without large effort and the authors suggest that it can also find applications beyond zebrafish embryos. Hence, it is not only interesting to the zebrafish community, but to a broader developmental biology audience.

Weakness of the study:

The main weakness of the manuscript is in the training data used for the deep-learning model and the apparent large impact of heterogeneous background illumination. If that is not solved, it is unclear if this technique will find an application in the zebrafish community.

We believe this weakness has now been addressed by incorporating additional data augmentation measures in the training process and testing the model on newly-acquired data.

October 3, 2023

RE: Life Science Alliance Manuscript #LSA-2023-02351

Dr. David J Barry
The Francis Crick Institute
1 Midland Road
London NW1 1AT
United Kingdom

Dear Dr. Barry,

Thank you for submitting your revised manuscript entitled "Automated staging of zebrafish embryos with deep learning". We would be happy to publish your paper in Life Science Alliance pending final revisions necessary to meet our formatting guidelines.

- please address Reviewer 1's remaining comment regarding image reflection
- please address Reviewer 3's minor comments
- please add an Author Contributions section to your main manuscript text and the system
- please upload your main manuscript text as an editable doc file
- please upload your figures as single files
- please add a Running Title and a Summary Blurb/Alternate Abstract to our system
- please add a Category for your manuscript in our system
- please add the Twitter handle of your host institute/organization as well as your own or/and one of the authors in our system
- please consult our manuscript preparation guidelines <https://www.life-science-alliance.org/manuscript-prep> and make sure your manuscript sections are in the correct order
- please incorporate any points from the Conclusion section into the Discussion; we only allow a Discussion section
- please use the [10 author names et al.] format in your references (i.e., limit the author names to the first 10)
- please upload your Tables in editable .doc or excel format;
- we encourage you to revise the figure legend for Figure 5 such that the figure panels are introduced in an alphabetical order
- please add callouts for Figures 6A-I and 7A-I to your main manuscript text

A. FINAL FILES:

B. MANUSCRIPT ORGANIZATION AND FORMATTING:

Sincerely,

Reviewer #1 (Comments to the Authors (Required)):

Reviewer #1 (Evidence, reproducibility, and clarity (Required)): In this manuscript the author is presenting a deep-learning model used to predict the development stage of zebrafish embryo. A robust method that can accurately classify a zebrafish into different development stages is highly relevant for many researchers working with zebrafish and hence the importance in developing methods like this is high.

The manuscript is overall ok. However, more data is needed to convince the reader that the method is robust enough to work with other samples in other labs. This would greatly improve the impact of the publication.

--We agree with the reviewer and have included in our revised manuscripts additional test data that was acquired at a different laboratory to the training data (Figures 5 - 7).

--This is great and provides the reader with more results on the robustness of the method and how it would perform on data acquired at different sites.

Page 6.-
How is the data acquired?

--Images used to do initial model training are the same as those used in a previous study - the details of image acquisition are contained in the relevant publication (doi: 10.12688/wellcomeopenres.18313.1). However, we have now added "Zebrafish Husbandry" and "Live Imaging" for newly-acquired images. We have added a table (Table 1) listing details of all image data used in the study.

--Thanks for the addition.

Page 8.

"This indicates that while KimmelNet can be used successfully with noisier test data than that on which it was trained, there is an upper limit to how noisy the data can be."

This is an obvious statement there will always be an upper limit on noise.

-We agree with the reviewer that this statement is not terribly informative. This section ("KimmelNet's prediction accuracy is not significantly impacted by moderate levels of additive noise") has been removed from the revised manuscript in favour of incorporating a section detailing testing of the model on newly-acquired images ("KimmelNet can generalise to previously unseen data").

--Great!

Page 9.

Are only wildtype embryos used? How would this work on different mutants. To evaluate the robustness of the method this it would be valuable to test on some mutant line with known developmental difference from the wild type.

-We agree with the reviewer that testing on a mutant line would lend more weight to our findings. For example, the p53^{-/-} zebrafish has a reported, published developmental delay, but we did not have access to that line. However, the developmental delay reported for the p53^{-/-} mutant is virtually indistinguishable from that effected by a temperature change. We therefore focussed our efforts on developmental delay affected by altering incubation temperature only.

--It would have been great to see data from a different mutant, but I see your reasoning and this seems like a good compromise.

Image data.

I would strongly suggest that the author should include a description of the data in the manuscript. A description of how the data is acquired, microscope, different batches, type of embryos used.

-The image data used in the first draft of the manuscript is the same as that used in a previous publication (Jones et al. 2022), which contains all the relevant details the reviewer seeks. However, we have now added the relevant information for the newly-acquired image data.

--Thanks this makes it easier for the reader to know the data and compare to others.

"Random translation in the y-direction was avoided due to the aspect ratio of the images (width>161height) - any artifacts introduced by translation in the x-direction would be removed by the 162centre crop layer, but this would not be the case for translation in the y-direction."

- Could this be solved by using some border method reflection, repetition or fixed value?"

-The reviewer is correct that some form of image reflection or repetition could be utilised. However, given the nature of our images, if an embryo is located close to the image boundary, reflection/repetition can result in some odd artefacts, so we minimised the use of such fill methods (also used by the random zoom augmentation layer). We could instead use an arbitrary fixed value, as the reviewer suggested, but finding a value suitable for all images is difficult.

--I see your point. However, when looking at the images in figure 2A I see no reason why translation in x would be more favourable than translation in y? Figure 2A row 4 column 4 this image is in the corner and would be equally affected by x and y translation. No need to modify just clarify better in text why you choose to do it this way.

Page 10.

Addition of Noise to Image Data

This should be added in the training phase. This would probably improve the robustness of the network and also improve the results on the test data.

-We agree with the reviewer and have now added a random Gaussian noise layer for data augmentation purposes during model training (see Figure 1).

--Great!

Supplementary 3 images with high noise. It is worrying that the network is not able to handle the noise in this figure. Looks like the features that is used to distinguish the developmental stage of the embryo is still clearly seen with this high noise level?

Retrain the model with noise as an augmentation to improve this.

--Ok good.

Reviewer #1 (Significance (Required)):

The development of methods like this is highly relevant in the zebrafish community. Staging and evaluating the developmental stage for zebrafish is common and is of interest to the broad community. A lot of this work today is done manually, limiting the

throughput, and adding human bias. The limit of this study is the dataset used for training and evaluation. Firstly, it is not clear about the structure of the data and how it is acquired, different types of fish or imaging setup etc. For a method to be useful to the community it needs to be robust enough to handle different types of fish (transgenic lines). The manuscript would be greatly improved by adding this to the training and evaluation.

-We have now added additional datasets for the purposes of evaluating the model.

--This new version of the manuscript is an improvement and better describes the method and shows how it will perform on other data than was used for training. I like the idea of fine-tuning on data acquired from a different batch and different microscope. Most dataset used with the method will be acquired in a different lab with different microscope so it is expected to have images that could look widely different. By using transfer learning this would address this issue and enable using KimmelNet on data acquired in a different lab possible without training the original version of KimmelNet on data from a large variety of microscopes and sites. The result on the new data after transfer learning is still quite spread and could be improved in the future. However, it is still possible to distinguish developmental delays between two groups which is the aim of the manuscript.

Reviewer #2 (Comments to the Authors (Required)):

The revised version of "Automated staging of zebrafish embryos with deep learning" by Barry et al., presents a method to automatically stage developmental timepoints of zebrafish embryos based on convolutional neural networks (CNN). My previous concerns with the paper have been sufficiently addressed by the authors. I would also like to compliment the authors on clearly stating what both the advantages and limitations of KimmelNet are which will make it easier for members of the zebrafish community to decide whether this is a useful tool for their particular application or not.

Reviewer #3 (Comments to the Authors (Required)):

1. EVIDENCE, REPRODUCIBILITY AND CLARITY

=====

In this study, the authors introduce KimmelNet, a deep learning model designed to predict the age of zebrafish embryo populations from 2D brightfield images. Addressing the challenge of accurately staging zebrafish embryo populations, KimmelNet offers robust performance even in the presence of measurement noise and demonstrates the ability to detect developmental delays using minimal experimental data.

Main results of the paper are:

- Development of a computational tool called Kimmelnet that allows for the staging of zebrafish embryos from microscopy images
- Usage of Kimmelnet to detect temperature-dependent developmental delay
- Successfully challenging Kimmelnet with unseen data acquired at a different experimental site

All main points are strongly supported by the provided results.

Overall, Kimmelnet is a clear advance in measuring delays in the development of zebrafish populations and promises that a similar network architecture and framework can also be used for other biological systems.

The study is well done, clearly written and code availability and added transparency of the used data ensure reproducibility.

Major comments:

All major comments have been sufficiently addressed.

Minor comments:

All minor comments of the previous review round have been sufficiently addressed except one:

- The authors did not correct the units in Eq. 1. It should have the form

$$H = h / (0.055 T - 0.57 K)$$

where K is Kelvin. I can also accept it if the authors write

$$H = h / (0.055 T - 0.57 \{\text{degree sign}\}C)$$

even though this is unusual. In any case, the authors must specify if T is given in Celsius or Kelvin.

Additional minor comments:

- Scalebars in Figure 2,4,5 missing.

Reviewer #1 (Comments to the Authors (Required)):

Reviewer #1 (Evidence, reproducibility, and clarity (Required)): In this manuscript the author is presenting a deep-learning model used to predict the development stage of zebrafish embryo. A robust method that can accurately classify a zebrafish into different development stages is highly relevant for many researchers working with zebrafish and hence the importance in developing methods like this is high.

The manuscript is overall ok. However, more data is needed to convince the reader that the method is robust enough to work with other samples in other labs. This would greatly improve the impact of the publication.

-We agree with the reviewer and have included in our revised manuscripts additional test data that was acquired at a different laboratory to the training data (Figures 5 - 7).

--This is great and provides the reader with more results on the robustness of the method and how it would perform on data acquired at different sites.

Page 6.-

How is the data acquired?

-Images used to do initial model training are the same as those used in a previous study - the details of image acquisition are contained in the relevant publication (doi: 10.12688/wellcomeopenres.18313.1). However, we have now added "Zebrafish Husbandry" and "Live Imaging" for newly-acquired images. We have added a table (Table 1) listing details of all image data used in the study.

--Thanks for the addition.

Page 8.

"This indicates that while KimmelNet can be used successfully with noisier test data than that on which it was trained, there is an upper limit to how noisy the data can be."

This is an obvious statement there will always be an upper limit on noise.

-We agree with the reviewer that this statement is not terribly informative. This section ("KimmelNet's prediction accuracy is not significantly impacted by moderate levels of additive noise") has been removed from the revised manuscript in favour of incorporating a section detailing testing of the model on newly-acquired images ("KimmelNet can generalise to previously unseen data").

--Great!

Page 9.

Are only wildtype embryos used? How would this work on different mutants. To evaluate the robustness of the method this it would be valuable to test on some mutant line with known developmental difference from the wild type.

-We agree with the reviewer that testing on a mutant line would lend more weight to our findings. For example, the p53^{-/-} zebrafish has a reported, published developmental delay, but we did not have access to that line. However, the developmental delay reported for the p53^{-/-} mutant is virtually indistinguishable from that effected by a temperature change. We therefore focussed our efforts on developmental delay affected by altering incubation temperature only.

--It would have been great to see data from a different mutant, but I see your reasoning and this seems like a good compromise.

Image data.

I would strongly suggest that the author should include a description of the data in the manuscript. A description of how the data is acquired, microscope, different batches, type of embryos used.

-The image data used in the first draft of the manuscript is the same as that used in a previous publication (Jones et al. 2022), which contains all the relevant details the reviewer seeks. However, we have now added the relevant information for the newly-acquired image data.

--Thanks this makes it easier for the reader to know the data and compare to others.

"Random translation in the y-direction was avoided due to the aspect ratio of the images (width>height) - any artifacts introduced by translation in the x-direction would be removed by the centre crop layer, but this would not be the case for translation in the y-direction."

- Could this be solved by using some border method reflection, repetition or fixed value?"

-The reviewer is correct that some form of image reflection or repetition could be utilised. However, given the nature of our images, if an embryo is located close to the image boundary, reflection/repetition can result in some odd artefacts, so we minimised the use of such fill methods (also used by the random zoom augmentation layer). We could instead use an arbitrary fixed value, as the reviewer suggested, but finding a value suitable for all images is difficult.

--I see your point. However, when looking at the images in figure 2A I see no reason why translation in x would be more favourable than translation in y? Figure 2A row 4 column 4 this image is in the corner and would be equally affected by x and y translation. No need to modify just clarify better in text why you choose to do it this way.

We have clarified this point in the text by modifying the last sentence in the Data Partitioning and Augmentation subsection as follows:

"Random translation in the y-direction was avoided due to the aspect ratio of the images (width > height) – due to the centre-crop layer only cropping in x (but not y), any artifacts introduced by translation in the x-direction would be largely removed, but those introduced by translation in y would not."

Page 10.

Addition of Noise to Image Data

This should be added in the training phase. This would probably improve the robustness of the network and also improve the results on the test data.

-We agree with the reviewer and have now added a random Gaussian noise layer for data augmentation purposes during model training (see Figure 1).

--Great!

Supplementary 3 images with high noise. It is worrying that the network is not able to handle the noise in this figure. Looks like the features that is used to distinguish the developmental stage of the embryo is still clearly seen with this high noise level? Retrain the model with noise as an augmentation to improve this.

--Ok good.

Reviewer #1 (Significance (Required)):

The development of methods like this is highly relevant in the zebrafish community. Staging and evaluating the developmental stage for zebrafish is common and is of interest to the broad community. A lot of this work today is done manually, limiting the throughput, and adding human bias. The limit of this study is the dataset used for training and evaluation. Firstly, it is not clear about the structure of the data and how it is acquired, different types of fish or imaging setup etc. For a method to be useful to the community it needs to be robust enough to handle different types of fish (transgenic lines). The manuscript would be greatly improved by adding this to the training and evaluation.

-We have now added additional datasets for the purposes of evaluating the model.

--This new version of the manuscript is an improvement and better describes the method and shows how it will perform on other data than was used for training. I like the idea of fine-tuning on data acquired from a different batch and different microscope. Most dataset used with the method will be acquired in a different lab with different microscope so it is expected to have images that could look widely different. By using transfer learning this would address this issue and enable using KimmelNet on data acquired in a different lab possible without training the original version of KimmelNet on data from a large variety of microscopes and sites. The result on the new data after transfer learning is still quite spread and could be improved in the future. However, it is still possible to distinguish developmental delays between two groups which is the aim of the manuscript.

Reviewer #2 (Comments to the Authors (Required)):

The revised version of "Automated staging of zebrafish embryos with deep learning" by Barry et al., presents a method to automatically stage developmental timepoints of zebrafish embryos based on convolutional neural networks (CNN). My previous concerns with the paper have been sufficiently addressed by the authors. I would also like to compliment the authors on clearly stating what both the advantages and limitations of KimmelNet are which will make it easier for members of the zebrafish community to decide whether this is a useful tool for their particular application or not.

Reviewer #3 (Comments to the Authors (Required)):

1. EVIDENCE, REPRODUCIBILITY AND CLARITY

=====

In this study, the authors introduce KimmelNet, a deep learning model designed to predict the age of zebrafish embryo populations from 2D brightfield images. Addressing the challenge of accurately staging zebrafish embryo populations, KimmelNet offers robust performance even in the presence of measurement noise and demonstrates the ability to detect developmental delays using minimal experimental data.

Main results of the paper are:

- Development of a computational tool called Kimmelnet that allows for the staging of zebrafish embryos from microscopy images
- Usage of Kimmelnet to detect temperature-dependent developmental delay
- Successfully challenging Kimmelnet with unseen data acquired at a different experimental site

All main points are strongly supported by the provided results.

Overall, Kimmelnet is a clear advance in measuring delays in the development of zebrafish populations and promises that a similar network architecture and framework can also be used for other biological systems.

The study is well done, clearly written and code availability and added transparency of the used data ensure reproducibility.

Major comments:

All major comments have been sufficiently addressed.

Minor comments:

All minor comments of the previous review round have been sufficiently addressed except one:

- The authors did not correct the units in Eq. 1. It should have the form

$$H = h / (0.055 T - 0.57 K)$$

where K is Kelvin. I can also accept it if the authors write

$$H = h / (0.055 T - 0.57 \text{ {degree sign}C})$$

even though this is unusual. In any case, the authors must specify if T is given in Celsius or Kelvin.

Apologies for this omission. We have modified the relevant section of the text as follows to make absolutely clear what units are used:

We used an equation of this form in order to be consistent with that used by Kimmel et al. (1995):

$$H_T = \frac{h}{0.055T - c}$$

(1)

where H_T corresponds to hours of development at temperature T (°C), h denotes the number of hours required to reach the equivalent developmental stage at 28.5°C and c is a constant equal to 0.57°C.

Additional minor comments:

- Scalebars in Figure 2,4,5 missing.

Thanks to the reviewer for picking up on this and apologies for the omission - we have now added scale bars to all images.

October 18, 2023

RE: Life Science Alliance Manuscript #LSA-2023-02351R

Dr. David J Barry
The Francis Crick Institute
1 Midland Road
London NW1 1AT
United Kingdom

Dear Dr. Barry,

Thank you for submitting your Methods entitled "Automated staging of zebrafish embryos with deep learning". It is a pleasure to let you know that your manuscript is now accepted for publication in Life Science Alliance. Congratulations on this interesting work.

DISTRIBUTION OF MATERIALS:

Again, congratulations on a very nice paper. I hope you found the review process to be constructive and are pleased with how the manuscript was handled editorially. We look forward to future exciting submissions from your lab.

Sincerely,
